# Examining trends in substance use disorder capacity and service delivery by Health Resources and Services Administration-funded health centers: A time series regression analysis

**Nadereh Pourat**[1,2]*, **Brenna O'Masta**[1], **Xiao Chen**[1], **Connie Lu**[1], **Weihao Zhou**[1], **Marlon Daniel**[3], **Hank Hoang**[4], **Alek Sripipatana**[4]

**1** Center for Health Policy Research, University of California, Los Angeles (UCLA), Los Angeles, CA, United States of America, **2** Fielding School of Public Health, UCLA, Los Angeles, CA, United States of America, **3** Center for Behavioral Health Statistics and Quality, Substance Abuse and Mental Health Services Administration. All work related to this manuscript was completed as an employee of Bureau of Primary Health Care, Health Resources and Services Administration, Rockville, MD, United States of America, **4** Bureau of Primary Health Care, Health Resources and Services Administration, U.S. Department of Health and Human Services, Rockville, MD, United States of America

* pourat@ucla.edu

**Data Availability Statement:** All relevant data are within the paper and its Supporting Information files.

## Abstract

### Background

The opioid epidemic and subsequent mortality is a national concern in the U.S. The burden of this problem is disproportionately high among low-income and uninsured populations who are more likely to experience unmet need for substance use services. We assessed the impact of two Health Resources and Services Administration (HRSA) substance use disorder (SUD) service capacity grants on SUD staffing and service use in HRSA -funded health centers (HCs).

### Methods and findings

We conducted cross-sectional analyses of the Uniform Data System (UDS) from 2010 to 2017 to assess HC (n = 1,341) trends in capacity measured by supply of SUD and medication-assisted treatment (MAT) providers, utilization of SUD and MAT services, and panel size and visit ratio measured by the number of patients seen and visits delivered by SUD and MAT providers. We merged mortality and national survey data to incorporate SUD mortality and SUD treatment services availability, respectively. From 2010 to 2015, 20% of HC organizations had any SUD staff, had an average of one full-time equivalent SUD employee, and did not report an increase in SUD patients or SUD services. SUD capacity grew significantly in 2016 (43%) and 2017 (22%). MAT capacity growth was measured only in 2016 and 2017 and grew by 29% between those years. Receipt of both supplementary grants increased the probability of any SUD capacity by 35% (95% CI: 26%, 44%) and service use, but decreased the probability of SUD visit ratio by 680 visits (95% CI: -1,013, -347), compared to not receiving grants.

**Funding:** This article was funded by the U.S. Department of Health and Human Services, Health Resources and Services Administration (HRSA, https://www.hrsa.gov/) under HRSA Contract number HHSH250201300023I (NP). The views expressed in this article are solely the opinions of the authors and do not necessarily reflect the official policies of the U.S. Department of Health and Human Services or HRSA, nor does mention of the department or agency names imply endorsement by the U.S. Government.

**Competing interests:** The authors have read the journal's policy, and the authors of this manuscript have the following competing interests to declare: MD, HH, and AS are employees of the U.S. Government, U.S. Department of Health and Human Services, which funded this study. This does not alter our adherence to PLOS ONE policies on sharing data and materials.

**Abbreviations:** HC, health center; SUD, substance use disorder; MAT, medication-assisted treatment; HRSA, Health Resource and Services Administration; SASE, Substance Abuse Service Expansion; AIMS, Access Increases in Mental Health and Substance Abuse Services; UDS, Uniform Data System; FTE, full-time equivalent.

## Conclusions

The significant growth in HC specialized SUD capacity is likely due to supplemental SUD-specific HRSA grants and may vary by structure of grants. Expanding SUD capacity in HCs is an important step in increasing SUD access for low income and uninsured populations broadly and for patients of these organizations.

## Introduction

The opioid problem, identified as early as the 1990s, has escalated since 2010 with heroin use and since 2013 with synthetic opioids [1, 2]. The problem is now considered an epidemic with an estimated 20.8 million (7.8%) individuals ages 12 and older in the United States reported to have substance use disorder (SUD) in 2015 [3]. During the same year, the rates of opioid use disorders were 0.6% for heroin and 2.0% for prescription opioids [4]. From 2000 to 2014, overdose deaths involving prescription opioids and heroin increased 200% and overdose deaths from all opioid drugs increased 137% [5]. Opioid use disorders alone led to approximately 14.9 deaths, 296 emergency department visits, (both in 2017) and 225 hospitalizations per 100,000 individuals (in 2014) [6, 7]. In addition, there were 16.3 opioid-related deaths per 100,000 persons in 2015 [5]. The escalation in opioid use is a significant problem, but other SUDs including alcohol and methamphetamine are prevalent causes of morbidity and preventable mortality and require treatment capacity [3]. The proportion of those who report needing but not receiving SUD treatment overall is as high as 89% nationally [8]. Furthermore, primary care settings are well positioned and recommended to provide treatment for individuals with SUDs [9, 10].

There are disparities in opioid use disorders and access to SUD services. Low-income individuals have higher rates of opioids misuse and opioid use disorder than the general U.S. population [11]. Uninsured and Medicaid patients had significantly worse access to SUD and mental health services than Medicare and privately insured individuals [12]. The former group cited various barriers contributing to unmet need for SUD care, including not being able to afford SUD treatment, lack of availability of needed SUD services, and greater distance in geographic proximity to SUD services [13]. Furthermore, non-Hispanic African-Americans and Hispanic/Latinos were found to have greater barriers to accessing SUD treatment services than non-Hispanic Whites [14, 15]. Increasing access to SUD services in primary care settings for these populations is considered an important strategy to combat the nation's opioid epidemic [16].In particular, medication-assisted treatment (MAT) delivered by health providers in primary care settings is essential to expanding access to opioid use treatment [17, 18].

Health Resources and Services Administration (HRSA)-funded health centers (HC) deliver primary care to racially/ethnically diverse and low-income and uninsured patients and can play a strategic and significant role in expanding access to SUD and MAT services to the underserved [19]. In 2017, 1,373 HC organizations with 11,056 delivery sites provided comprehensive and affordable primary care to over 27 million, or 1 in 12 Americans, 63 percent of whom are racial/ethnic minorities [20, 21].

In a 2014 national survey of patients served by HRSA-funded HCs, patients reported a slightly lower risk of SUD and opioid use disorder compared to national rates, with 6.4% of HC patients at moderate or high risk of SUD and 1.2% were estimated to abuse opioids and/or were dependent on them, though it is likely that much of the available SUD data are self-reported and underestimate SUD risk [22–24]. In 2018, nearly 70 percent of HC organizations

provided SUD services in at least one delivery site, but on-site service provision was not universal as HCs can arrange for services to be provided through contracted providers or can refer patients to providers with informal agreements [18].

In March 2016, HRSA awarded $94 million through the Substance Abuse Service Expansion (SASE) supplemental grants to 271 or 20% of HCs, with supplementary funding ranging from $217,000 to $406,000. The grants were awarded to increase numbers of SUD providers and staff and increase access to SUD and MAT services [25]. HCs that received these grants were required to: 1) enhance or establish an integration primary care/behavioral health model; 2) increase patients screened through Screening, Brief, Intervention, and Referral to Treatment (SBIRT) model; and 3) increase access to patients by either adding at least 1.0 full-time equivalent SUD provider or by adding or enhancing existing SUD services; 4) coordinate services necessary for patients; and 5) provide training and educational resources to assist health professional in making informed prescribing decisions. In September 2017, HRSA awarded another $200 million through Access Increases in Mental Health and Substance Abuse Services (AIMS) supplemental grants to 1,178 or 86% of HCs, with grants ranging from $84,000 to $176,000 [26]. AIMS grants included supporting the expansion of SUD services with a focus on opioid abuse, but also sought to increase mental health services. Recipients of this grant were required to: 1) expand direct hire staff and/or contractors who will support mental health and substance abuse service expansion focusing on the treatment, prevention, and awareness of opioid abuse; 2) provide access to expanded mental health and substance abuse services; and 3) increase the number of mental health patients and/or substance abuse patients as a result of AIMS funding. SASE provided awards of up to $325,000 per year for two years (2016 through 2018), while AIMS awarded funding divided into one-time or ongoing supplements, of up to $75,000 per year.

This study seeks to understand the impact of these targeted funding mechanisms on capacity and ability to improve access to SUD services provided by specialized SUD staff. Several recent studies have examined behavioral health capacity at HCs and suggest gaps in current SUD capacity and service use at HCs [27–33]. In addition, a limited number of studies have directly examined the potential role of recent HRSA investments in SUD capacity and service use at HCs, with emerging evidence suggesting such investments were associated with addiction treatment capacity [34]. To address these gaps, we examined the trends in HCs in specialized SUD and MAT capacity, service use, and patients and visits per provider in general and the potential impact of the 2016 and 2017 HRSA grants on these indicators. We hypothesized that HRSA grants would increase the number of specialized SUD providers and staff who would deliver more visits to more patients. The results provide important information on SUD capacity in the primary care settings most commonly used by low-income and uninsured populations and further highlight how this capacity may be increased to address the opioid epidemic and the need for SUD services. Because SASE funding was distributed in March 2016 and AIMS funding was distributed in September 2017, we anticipated that the impact of SASE was discernible in 2016 and 2017, but the impact of AIMS would be discernible partly in 2017 with the full impact mostly observed in 2018 and subsequent years. We focused on specialized SUD staff because we could not measure SUD services provided by medical or mental health providers, other than MAT.

## Methods

### Data and sample

For this study, we used a time-series analysis using HC administrative data and publicly available data sources to assess the impact of HRSA supplemental funding on our outcomes. We

used the 2010 to 2017 Uniform Data System (UDS) to examine HC staffing and delivery of MAT and SUD services. UDS is an annual cross-sectional administrative database maintained by HRSA. All HRSA-funded HCs are required to submit a UDS report annually including data on patient characteristics, staffing, utilization of services, and revenues from the previous calendar year. UDS captures aggregate information at the HC organizational level (i.e., parent organization or network level) rather than individual delivery sites that operate within the organization. Each HC organization operates multiple delivery sites.

To compare the size of HC SUD and MAT staffing with national and regional need for these services, we obtained publicly available opioid-related mortality rates from Centers for Disease Control and Prevention (CDC) Wide-ranging OnLine Data for Epidemiologic Research (WONDER) [4, 35]. We merged the CDC WONDER database with UDS data using the Federal Information Process Standards (FIPS) code associated with the address of the HC organization. If the FIPS code matched to more than one county, the county with the larger share of HC patients was selected. We also used the 2015 National Survey of Substance Abuse Treatment Facilities (N-SSATS) to assess the supply of drug and alcohol treatment facilities in the county where the HC was located. We merged the 2015 N-SSATS with UDS using FIPS code. We included all 1,375 HCs in the descriptive analyses, but for models we excluded 21 HCs that were not operational or did not report data in 2015. We also excluded 13 HCs that only received SASE funding from the models because the small sample size led to unreliable results. Our final 2015 sample for the models was 1,341 HCs.

### Independent variables

The primary variable of interest was receipt of HRSA funding status. HRSA's Bureau of Primary Health Care provided the list of HCs that received SASE and AIMS awards. We categorized HCs into those that received (1) no grants, (2) AIMS only, and (3) both SASE and AIMS (SASE/AIMS) grants. We controlled for several HC and market characteristics in 2015. HC characteristics included the size of the HC organization, indicated by whether the HC served more than 10,000 patients (vs. fewer), and use of electronic health records as an indicator of capacity to incorporate population health management [36]. Other HC organizational characteristics included urban/rural status of the HC organization and U.S. Census regions. The latter included New England, Middle Atlantic, East North Central, West North Central, South Atlantic, East South Central, West South Central, Mountain, and Pacific regions. We also included several measures of case mix and socioeconomic profile of HC patients. These included the proportion of patients who were racial/ethnic minorities, homeless, agricultural workers, below 100% of federal poverty guidelines, uninsured, or had Medicaid coverage. We identified whether the HC achieved patient-centered medical home (PCMH) recognition in the UDS, as an indicator of a comprehensive and integrated approach to care. PCMH recognition is reported at the HC-organization level, although recognition can be given to specific sites as well as to the whole organization. We also controlled availability of SUD services by the average number of facilities providing substance abuse services per 100,000 persons by county, extracted from the 2015 N-SSATS, and controlled for the opioid mortality rate per 100,000 persons by state, acquired from CDC WONDER.

### Dependent variables

We measured specialized SUD capacity by (1) the proportion of HCs with at least one full-time equivalent (FTE) specialized SUD staff, (2) the average number of SUD staff per HC, and (3) the ratio of SUD staff per 1,000 patients. HCs report substance abuse workers, psychiatric or mental health nurses, psychiatric or clinical social workers, clinical psychologists, and other

individuals providing alcohol or drug abuse counseling and/or treatment services as SUD staff. We did not include mental health or primary care providers that could deliver SUD services because UDS did not require HCs to report these provider's provision of SUD services separately from medical or mental health services. We next measured SUD service use by (1) the proportion of FTE SUD staff to total HC patients and (2) the proportion of FTE SUD staff to total HC visits. We then measured panel size and visit ratio of SUD staff by calculating (1) the ratio of SUD patients per FTE SUD staff per year (panel size) and (2) the average number of SUD visits per FTE SUD staff. HCs report SUD visits provided by SUD staff and SUD patients as patients that had at least one SUD visit.

We also measured MAT capacity, service use, and provider panel size in 2016 and 2017. HCs reported providers with Drug Addiction Treatment Act of 2000 (DATA 2000) waivers, most of whom are likely to be primary care providers, for MAT starting in 2016. MAT allows providers to prescribe specific controlled medications for opioid dependency treatment, but the DATA waiver is not inclusive of all treatment possibilities (i.e., implantable formulations of naloxone/buprenorphine) [37]. Our measures of MAT capacity included the (1) proportion of HCs with MAT providers, (2) average number of MAT providers per HC, and (3) the ratio of MAT provider per 1,000 patients. We measured MAT service use by the proportion of total HC patients that were MAT patients. The UDS does not include the number of MAT visits. We measured MAT panel size by the ratio of MAT patients per MAT provider at each HC.

## Analytic methods

We assessed the changes in our outcomes of interest from 2010 through 2017 for all HCs in the UDS in these years. We calculated the average annual percent change in SUD capacity, service use, and panel size and visit ratio from 2010 to 2015 to establish trends prior to 2016. We then calculated the percent change from 2015 to 2016 and 2016 to 2017 separately to assess increases or decreases since 2015 on our outcomes. We only measured changes in MAT capacity, service use, and panel size between 2016 and 2017 descriptively and conducted chi-square or t-tests as appropriate. We then constructed random-effects logistic, negative binomial, or Poisson regression models, as appropriate, controlling for HC patient and organizational characteristics and other likely confounders to assess the potential role of HRSA funding on outcomes. We included only complete data for all analyses presented in this paper. All analyses were conducted using Stata v.15 and Margins post-estimation command to report predicted probabilities for ease of interpretation. All statistically significant results with probability values of 0.05 or smaller were discussed.

## Ethics statement

This research was granted exemption by the University of California Los Angeles Instructional Review Board (study number 16–001528) due to secondary analysis of de-identified and publicly available data.

## Results

The national opioid-mortality rate increased at an average annual percent change of 11% from 2010 to 2015. This rate grew by 28% in 2016 and 12% in 2017 (Table 1). While the number of HCs grew from 1,124 in 2010 to 1,373 in 2017, the proportion of HCs with any SUD staff declined on average 1% per year from 2010 to 2015 but increased by 43% from 2015 to 2016 and 22% from 2016 to 2017. By 2017, more than one-third (35%) of HCs had any FTE SUD staff. The average number of SUD staff declined by 2% from 2010 to 2015 but increased by 21% in 2016 and by 17% in 2017. This trend was consistent with the ratio of SUD staff for

**Table 1. Provision of Substance Use Disorder (SUD) and Medication-Assisted Treatment (MAT) services by federally-funded Health Centers (HCs) in the United States in 2010 and 2015 to 2017[a,b].**

| | 2010 | 2015 | 2016 | 2017 | Average Annual Percent Change between 2010 to 2015[c] | Annual Percent Change 2015 to 2016 | Annual Percent Change 2016 to 2017 |
|---|---|---|---|---|---|---|---|
| Number of HCs | 1,124 | 1,375 | 1,367 | 1,373 | 4% | -1% | 0.4% |
| | *Mean (SD) or % (n)* | *Mean (SD) or % (n)* | *Mean (SD) or % (n)* | *Mean (SD) or % (n)* | | | |
| *SUD Capacity* | | | | | | | |
| Proportion of health centers with FTE SUD staff | 20% (230) | 20% (274) | 28% (388) | 35% (477) | -1% | 43% | 22% |
| Average number of FTE SUD staff | 1.0 (4.8) | 0.9 (4.6) | 1.2 (4.7) | 1.4 (4.9) | -2% | 21% | 17% |
| Average FTE SUD staff per 1,000 patients | 0.11 (0.68) | 0.10 (0.58) | 0.11 (0.69) | 0.12 (0.55) | -1% | 13% | 4% |
| *SUD Service Use* | | | | | | | |
| Average proportion of total HC patients that were SUD patients | 1.1% (3.8%) | 1.2% (5.0%) | 1.1% (3.2%) | 1.2% (3.3%) | 1% | -10% | 11% |
| Average proportion of total HC visits that were SUD visits | 1.7% (5.5%) | 1.4% (4.7%) | 1.4% (4.5%) | 1.4% (3.8%) | -4% | 5% | 1% |
| *SUD Panel Size and Visit Ratio* | | | | | | | |
| Average SUD patients per FTE SUD staff | 318 (472) | 249 (277) | 295 (1,073) | 264 (904) | -4% | 19% | -10% |
| Average SUD visit per FTE SUD staff | 1,580 (4,185) | 1,128 (1,492) | 1,263 (3,460) | 1,129 (3,029) | -6% | 12% | -11% |
| *MAT Capacity* | | | | | | | |
| Proportion of health centers with MAT providers | —[d] | — | 33% (453) | 43% (588)[§§§ e] | — | — | 29% |
| Average number of MAT providers | — | — | 1.2 (3.8) | 2.2 (5.6)[§§§] | — | — | 73% |
| Average MAT providers per 1,000 patients | — | — | 0.12 (0.7) | 0.17 (0.5) | — | — | 37% |
| *MAT Service Use* | | | | | | | |
| Average proportion of total HC patients that were MAT patients | — | — | 0.2% (1.0%) | 0.4% (1.1%)[§§ e] | — | — | 52% |
| *MAT Panel Size* | | | | | | | |
| Average MAT patients per MAT provider | — | — | 28.7 (44.8) | 26.3 (40.1) | — | — | -8% |
| *National Opioid-Mortality Rate* | | | | | | | |
| National Opioid-mortality rate per 100,000 persons | 6.8 (0.05) | 10.4 (0.06) | 13.3 (0.07) | 14.9 (0.07) | 11% | 28% | 12% |

[a] SUD = substance use disorder, MAT = medication assisted treatment; includes drugs and alcohol.

[b] Authors' analyses of data from the 2010 to 2017 Uniform Data System.

[c] The average percent change for 2010–2011, 2011–2012, 2012–2013, 2013–2014 and 2014–2015.

[d] MAT services were reported in the Uniform Data System starting in 2016.

[e] Statistically significant comparing 2017 to 2016 at §p<0.05,

§§p<0.01,

§§§p<0.001.

Standard deviation or count in parentheses.

SUD, substance use disorder; MAT, medication assisted treatment; SD, standard deviation; HC, health center.

1,000 patients per HC. The proportion of SUD to total HC patients grew 1% on average from 2010 to 2015, declined 10% in 2016, and grew 11% in 2017. The proportion of SUD to total

HC visits declined 4% on average from 2010 to 2015, but then grew 5% in 2016 and 1% in 2017. SUD panel size or the ratio of SUD patient to SUD staff declined by 4% from 2010 to 2015, increased by 19% in 2016, and declined by 10% in 2017. The same trends were observed for the ratio of SUD visits per SUD staff.

The proportion of HCs with MAT providers increased by 29% from 2016 to 2017. Similarly, the average number of MAT providers increased 73% from 1.2 MAT providers in 2016 to 2.2 in 2017. From 2016 to 2017, MAT service use as measured by the proportion of MAT patients to total HC patients grew 52%. MAT panel size decreased by 8% over the same time period.

The majority (68%) of HCs received an AIMS grant only, 19% received both AIMS and SASE, and 13% received neither (Table 2). Examining the SUD staff capacity of HCs in 2015 prior to grant distributions showed that those with both SASE/AIMS funding and those with AIMS only were more likely to have any SUD staff and a higher average number of SUD staff per HC than those without. However, the ratio of SUD staff per 1,000 patients, the ratio of SUD patients per SUD staff, and the ratio of SUD visits per SUD staff was not different by funding. Examining SUD service use showed that SASE/AIMS HCs were more likely to have a higher proportion of SUD visits, but the proportion of SUD patients did not differ by funding.

Examining HC characteristics by funding status showed that those with supplemental funding were larger, located in the Pacific and New England census regions, and more likely to have electronic health records or PCMH recognition than those without funding. These HCs also had more patients that were homeless, uninsured, or Medicaid beneficiaries compared to HCs without funding. Similarly, these HCs were located in counties that had a higher average number of substance abuse facilities and a higher opioid mortality rate per 100,000 persons. HCs with SASE/AIMS funding were also more likely to have any SUD staff and be located in East North Central and New England census regions than those with AIMS only funding. Those with AIMS only funding were also more likely to be located in urban areas than those without funding.

Table 3 displays measures of SUD staffing, service use, panel size, and visit ratio from 2015 to 2017 and percent change in those measures during that same time period given HRSA funding status and after controlling for HC and county characteristics. Among HCs without funding, there was no increase or primarily a decline in staffing, service use, and panel size but an increase in visits per SUD staff from 2015 to 2017. However, HCs with SASE/AIMS funding were more likely to have added SUD staff (115%), more SUD staff (18%), and more SUD staff per 1,000 patients (63%) from 2015 to 2017. HCs with AIMS only showed a 49% growth in SUD staff and limited growth in other SUD capacity measures from 2015 to 2017. These HCs also showed a percentage change decline in service use. Regardless of funding status, all HCs saw a percentage change decrease in SUD panel size.

Table 4 presents the predicted probabilities for SUD capacity, service use, panel size, and visit ratio based on multivariate models controlling for covariates. The percentage change increase in SUD capacity among HCs with SASE/AIMS corresponded to significant predictive probability increases in service use indicators compared to HCs with AIMS only or no funding. For example, HCs with SASE/AIMS funding increased the probability of adding SUD staff by 26% (95% CI: 18%, 33%), the number of SUD staff by 0.4 FTE (95% CI: 0.2, 0.7), and more SUD staff per 1,000 patients by 0.10 FTE (95% CI: 0.05, 0.15), compared to HCs with AIMS only funding. Similar trends were seen when comparing predicted probabilities between HCs with SASE/AIMS funding to HCs without funding. When compared to HCs without funding, HCs with AIMS funding only increased the probability of hiring SUD staff by 10% (95% CI: 3%, 16%) and had no impact on other measures of SUD capacity. HCs with AIMS only funding decreased the probability of the proportion of SUD visits by 0.3% (95% CI: -0.4%, -0.2%) compared to HCs with no funding. However, HCs with both SASE/AIMS increased the

**Table 2. Health center characteristics by SASE and AIMS grantee status in 2015.**

| | 2015 (Baseline) | | | | |
| --- | --- | --- | --- | --- | --- |
| | **Total** | **None** | **AIMS Only** | **SASE/AIMS** | **p-value** |
| Number of HCs | 1,341 | 13% (176) | 68% (907) | 19% (258) | |
| | *Mean (SD) or % (n)* | *Mean (SD) or % (n)* | *Mean (SD) or % (n)* | *Mean (SD) or % (n)* | |
| ***SUD Capacity*** | | | | | |
| Proportion of health centers with SUD staff | 20% (269) | 10% (18) | 17% (153) | 38% (98) | 0.000 |
| Average number of SUD staff | 1.0 (4.7) | 0.6 (3.4) | 0.8 (4.3) | 1.9 (6.2) | 0.010 |
| Average SUD staff per 1,000 patients | 0.1 (0.6) | 0.1 (0.7) | 0.1 (0.6) | 0.1 (0.3) | 0.870 |
| ***SUD Panel Size and Visit Ratio*** | | | | | |
| Average SUD patients per SUD staff | 248.3 (278.0) | 266.2 (313.3) | 220.1 (250.4) | 288.6 (308.1) | 0.160 |
| Average SUD visit per SUD staff | 1123.6 (1497.4) | 949.5 (923.2) | 1039.9 (931.6) | 1283.9 (2144.9) | 0.401 |
| ***SUD Service Use*** | | | | | |
| Average proportion of total HC patients that were SUD patients | 1.2% (5.1%) | 0.9% (3.3%) | 1.0% (5.7%) | 1.8% (3.6%) | 0.139 |
| Average proportion of total HC visits that were SUD visits | 1.4% (4.7%) | 0.9% (3.4%) | 1.2% (4.7%) | 2.3% (5.5%) | 0.006 |
| ***Health Center Characteristics*** | | | | | |
| More than 10,000 patients served last year | 53.7% (720) | 73.6% (80) | 49.6% (450) | 73.6% (190) | 0.000 |
| Urban (vs. rural) | 44.8% (601) | 50.4% (71) | 44.1% (400) | 50.4% (130) | 0.071 |
| **Region** | | | | | |
| New England | 7.7% (103) | 15.4% (3) | 6.6% (60) | 15.4% (40) | 0.000 |
| Middle Atlantic | 9.8% (132) | 10.6% (16) | 9.8% (89) | 10.6% (27) | |
| East North Central | 12.6% (169) | 18.5% (13) | 11.9% (108) | 18.5% (48) | |
| West North Central | 6.9% (93) | 5.5% (7) | 7.9% (71) | 5.5% (14) | |
| South Atlantic | 16.7% (224) | 13.4% (34) | 17.2% (156) | 13.4% (35) | |
| East South Central | 6.6% (88) | 3.5% (16) | 7.0% (63) | 3.5% (9) | |
| West South Central | 10.5% (140) | 3.5% (33) | 10.9% (99) | 3.5% (9) | |
| Mountain | 8.5% (114) | 7.1% (10) | 9.4% (86) | 7.1% (18) | |
| Pacific | 20.8% (278) | 22.4% (45) | 19.4% (176) | 22.4% (58) | |
| Had Electronic Health Records | 72.8% (976) | 82.6% (105) | 72.6% (658) | 82.6% (213) | 0.000 |
| Percent with PCMH Recognition | 68.4% (917) | 85.7% (88) | 67.0% (608) | 85.7% (221) | 0.000 |
| ***Health Center Patient Characteristics*** | | | | | |
| Percent of Patients that are Minority | 55.6% (32.0%) | 55.2% (33.9%) | 55.0% (32.1%) | 57.9% (30.6%) | 0.424 |
| Percent of Patients that are Homeless | 7.3% (19.2%) | 5.7% (18.3%) | 6.5% (18.0%) | 11.1% (23.1%) | 0.002 |
| Percent of Patients that are Migrant and Agricultural Workers | 2.8% (10.5%) | 3.4% (13.9%) | 2.7% (10.5%) | 2.5% (7.5%) | 0.606 |
| Percent of Patients that are less than 100% of the Federal Poverty Guideline | 48.4% (24.4%) | 46.3% (24.7%) | 48.1% (24.2%) | 50.7% (25.1%) | 0.153 |
| Percent of Patients that are Uninsured | 27.0% (19.3%) | 28.6% (21.5%) | 27.9% (19.5%) | 23.0% (16.1%) | 0.001 |
| Percent of Patients that are Medicaid | 43.1% (19.9%) | 39.8% (21.1%) | 41.8% (19.9%) | 50.0% (17.4%) | 0.000 |
| ***County Characteristics*** | | | | | |
| Mean number of facilities per 100,000 persons providing substance abuse services | 4.1 (2.1) | 3.8 (2.3) | 4.0 (2.1) | 4.3 (2.0) | 0.042 |
| Opioid mortality rate per 100,000 persons | 10.9 (6.4) | 10.1 (6.2) | 10.6 (6.2) | 12.7 (6.8) | 0.000 |

Standard deviation or count in parentheses. Analyses involved comparing independent and control variables by supplemental funding status using t-test or chi-square test, as appropriate.

SASE, Substance Abuse Service Expansion; AIMS, Access Increases in Mental Health and Substance Abuse Services; HC, health center; PCMH, patient centered medical home; SUD, substance use disorder.

probability of number of patients per SUD staff by 234 patients (95% CI: 16, 452) while HCs with AIMS only increased the probability by 314 patients (95% CI: 97, 532), compared to HCs

**Table 3. Adjusted provision of Substance Use Disorder (SUD) services by federally-funded health centers in the United States between the year prior to grant distribution (2015) and the last year the grants were distributed (2017), by grantee status.**

| | None | | | AIMS Only[b] | | | SASE/AIMS[a,b] | | | Percent Change 2015–2017 | | |
|---|---|---|---|---|---|---|---|---|---|---|---|---|
| | 2015 | 2016 | 2017 | 2015 | 2016 | 2017 | 2015 | 2016 | 2017 | None | AIMS only | SASE/AIMS |
| Number of HCs | 13% (176) | | | 68% (907) | | | 19% (258) | | | | | |
| | *Mean (SD) or % (n)* | | | *Mean (SD) or % (n)* | | | *Mean (SD) or % (n)* | | | | | |
| ***SUD Capacity*** | | | | | | | | | | | | |
| Proportion of health centers with SUD personnel | 11% (19) | 12% (21) | 10% (17) | 17% (157) | 19% (174) | 26% (235) | 35% (91) | 68% (175) | 76% (195) | -10% | 49% | 115% |
| Average number of SUD staff | 1.4 (0.4) | 1.1 (0.4) | 1.3 (0.4) | 1.9 (0.3) | 2.0 (0.3) | 2.0 (0.3) | 3.0 (0.3) | 3.3 (0.3) | 3.6 (0.3) | -6% | 6% | 18% |
| Average SUD staff per 1,000 patients | 0.04 (0.01) | 0.04 (0.01) | 0.04 (0.01) | 0.07 (0.01) | 0.07 (0.01) | 0.07 (0.01) | 0.16 (0.03) | 0.20 (0.03) | 0.26 (0.04) | -9% | 1% | 63% |
| ***SUD Service Use*** | | | | | | | | | | | | |
| Average proportion of total HC patients that were SUD patients | 1.5% (0.5%) | 0.9% (0.4%) | 1.1% (0.4%) | 2.0% (0.4%) | 2.0% (0.4%) | 2.0% (0.4%) | 5.0% (1.4%) | 5.3% (1.5%) | 5.8% (1.7%) | -22% | -1% | 16% |
| Average proportion of total HC visits that were SUD visits | 0.7% (0.3%) | 0.6% (0.2%) | 0.7% (0.3%) | 2.1% (0.4%) | 2.1% (0.4%) | 1.8% (0.3%) | 4.8% (1.4%) | 5.2% (1.5%) | 5.7% (1.7%) | -1% | -13% | 18% |
| ***SUD Panel Size and Visit Ratio*** | | | | | | | | | | | | |
| Average SUD patients per SUD staff | 1,050 (323) | 818 (251) | 696 (214) | 342 (30) | 296 (26) | 302 (26) | 472 (45) | 392 (37) | 352 (33) | -34% | -13% | -26% |
| Average SUD visit per SUD staff | 1,152 (323) | 1673 (469) | 1755 (492) | 1081 (81) | 1101 (82) | 963 (72) | 1209 (104) | 1201 (103) | 1131 (97) | 52% | -11% | -6% |

[a] HRSA distributed $94 million in March 2016 through the Substance Abuse Service Expansion (SASE) program to 271 HCs ($217,000 to $406,000) with the goals of increasing SUD personnel, increasing number of patients screened and connected to SUD treatment, and increasing access to Medication Assisted Treatment (MAT) services.

[b] HRSA distributed $200 million in September 2017 through the Access Increases in Mental Health and Substance Abuse Services (AIMS) program to 1,178 HCs ($84,000 to $176,000) with the goals of increasing substance abuse services focusing on the treatment, prevention, and awareness of opioid abuse; increasing SUD personnel; and leveraging health information technology and training to increase and improve SUD services.

Standard deviation or count in parentheses.

SD, Standard Deviation; SUD, substance use disorder; AIMS, Access Increases in Mental Health and Substance Abuse Services; SASE, Substance Abuse Service Expansion; HC, health center; HRSA, Health Resources and Services Administration; MAT, Medication Assisted Treatment.

with no funding. Visits per SUD staff increased the probability among HCs with SASE/AIMS funding by 41 visits (95% CI: 18, 63) compared to those with HCs with AIMS only funding and decreased the probability by 680 visits (95% CI: -1,013, -347) compared to HCs with no funding. The same pattern was observed for HCs with AIMS only compared to those without funding, a probability decrease of 720 visits (95% CI: -1,054, -387). The final regression models are displayed in the S1 Table.

## Discussion

We did not find growth in specialized SUD capacity, service use, panel size or visit ratio prior to 2015. Annual percent change in all measures were significantly different after 2015 from prior, with growth in SUD capacity and service use but a decline in provider panel size and visits. Regression findings indicate changes from 2015 to 2017 seemed to be in response to SASE/AIMS funding efforts. The funding seemed to have promoted more HCs to hire SUD staff, and those with existing staff to hire more of such providers. Increased staffing corresponded to increased service delivery but produced fewer SUD visits compared to HCs that did not receive funding. Changes in MAT capacity, panel size, and service use were only measurable from

**Table 4. Predicted probabilities of Substance Use Disorder (SUD) services by federally-funded health centers in the United States between the year prior to grant distribution (2015) and the last year the grants were distributed (2017), by grantee status.**

| | Predicted Probability [95% CI] | | |
|---|---|---|---|
| | AIMS only vs. none | SASE/AIMS vs. none | SASE/AIMS vs. AIMS only |
| Number of HCs | 13% (176) | 68% (907) | 19% (258) |
| **SUD Capacity** | | | |
| Proportion of health centers with SUD personnel | 10% [3%, 16%]*** | 35% [26%, 44%]*** | 26% [18%, 33%]*** |
| Average number of SUD staff | 0.2 [-0.3, 0.7] | 0.6 [0.1, 1.2]* | 0.4 [0.2, 0.7]*** |
| Average SUD staff per 1,000 patients | 0.00 [-0.02, 0.03] | 0.10 [0.05, 0.16]*** | 0.10 [0.05, 0.15]*** |
| **SUD Service Use** | | | |
| Average proportion of total HC patients that were SUD patients | 0.3% [0.1%, 0.5%]*** | 1.1% [0.6%, 1.7%]*** | 0.8% [0.4%, 1.3%]*** |
| Average proportion of total HC visits that were SUD visits | -0.3% [-0.4%, -0.2%]*** | 0.9% [0.4%, 1.4%]** | 1.1% [0.6%, 1.7%]*** |
| **SUD Panel Size and Visit Ratio** | | | |
| Average SUD patients per SUD staff | 314 [97, 532]** | 234 [16, 452]* | -80 [−104, −57]*** |
| Average SUD visit per SUD staff | -720 [−1,054, −387]*** | -680 [−1,13, −347]*** | 41 [18, 63]*** |

[a] HRSA distributed $94 million in March 2016 through the Substance Abuse Service Expansion (SASE) program to 271 HCs ($217,000 to $406,000) with the goals of increasing SUD personnel, increasing number of patients screened and connected to SUD treatment, and increasing access to Medication Assisted Treatment (MAT) services.
[b] HRSA distributed $200 million in September 2017 through the Access Increases in Mental Health and Substance Abuse Services (AIMS) program to 1,178 HCs ($84,000 to $176,000) with the goals of increasing substance abuse services focusing on the treatment, prevention, and awareness of opioid abuse; increasing SUD personnel; and leveraging health information technology and training to increase and improve SUD services.
Statistically significant at
*p<0.05;
**p<0.01;
***p<0.001.
SUD, substance use disorder; AIMS, Access Increases in Mental Health and Substance Abuse Services; SASE, Substance Abuse Service Expansion; HC, health center; CI, confidence interval.

2016 to 2017 and all measures except the ratio of MAT patients per MAT provider and the ratio of MAT providers per 1,000 patients increased significantly. However, the MAT patient-to-provider ratio (26.3 patients per provider) remains below the initial 30-patient limit set for providers [38].

The stagnant growth in specialized SUD capacity prior to 2015 was in contrast to the rise in national opioid overdose deaths since 2010 [1]. However, the rapid growth in SUD capacity among HCs with SUD funding by 2017 indicates the likely effectiveness of the targeted effort by HRSA to promote capacity. This is consistent with previous research that indicates when such investments are made to HCs, an increase in staffing is observed [39]. The growth in SUD capacity was timely and resulted in 0.11 SUD FTE staff per 1,000 patients in 2016, compared to national estimates of 0.07 outpatient SUD treatment staff per 1,000 patients [40]. Despite this higher level of capacity, the estimated level of need in HCs indicates that this capacity is likely to be inadequate [41]. Further considerations should be given in continuous supplemental funding (as opposed to one-time funding) and for future targeted technical

assistance to encourage providers to apply for patient limit waivers, further increasing MAT access for patients, and continue to remove cultural and societal stigmas associated with seeking SUD services [28, 42].

The combination of SASE/AIMS funding was associated with predicted probability increases in the proportion of patients and visits compared to receiving only AIMS funding or no funding. For example, the proportion of patients and visits that were associated with SUD services increased at HCs with SASE/AIMS compared to HCs receiving only AIMS funding or no funding. However, only the proportion of patients associated with SUD services increased at HCs with AIMS only compared to HCs without funding. Similarly, we found an increase in the number of SUD patients and visits for HCs with both SASE/AIMS compared to both HCs receiving AIMS only and an increase in SUD patients but a decrease in SUD visits for HCs with AIMS only versus HCs without funding (S2 Table). This difference may have been due to distribution of AIMS funds in September 2017 and our inability to assess the full impact of this grant mechanism. In other words, HCs with AIMS only funding experienced growth in SUD staffing and the number of SUD patients, but we did not capture growth in other measures of capacity and service use, potentially due to the limited maturity of the AIMS grant. It is possible HCs that received both SASE/AIMS were initially better equipped to expand SUD capacity and services, due to the requirements of SASE to establish an integrated behavioral health/primary care model, compared to HCs that received AIMS only. Those with AIMS only may not have been equipped or incentivized to sustain increased SUD capacity.

The growth in specialized SUD capacity corresponded to a decrease in productivity of SUD staff. As an expected result of supplemental funding, these improvements may have led to changes in practice patterns among HCs that received these grants, but the patient panel size for SUD providers is still substantially lower than other provider types [40]. While increased staffing may have reduced the burden of care on existing providers and permitted reductions in panel size and provision of visits, there appears to be a balance between managing patients with increasing complex conditions and addressing unmet SUD needs in communities [43].

The increase in MAT capacity and service use from 2016 to 2017 within HCs is noteworthy and highlights HC efforts to expand their ability to manage SUD patients by primary care providers. Together, increased co-location and expansion of SUD and MAT capacity in HCs highlight the progress towards on-site SUD service delivery in these primary care settings. This progress is likely to benefit HC patients and potentially combat the opioid epidemic and its consequences. Our findings indicate that room for developing additional capacity still exists, particularly at HCs without SUD and MAT capacity. Our findings also imply the importance of identifying solutions to sustaining this growth in the longer term.

Growth in specialized SUD capacity is dependent of availability of such workforce and ability of HCs to recruit and retain these providers. For example, a report to Congress indicated that rural areas have greater turnover and challenges in recruiting SUD providers, and these shortages have a direct impact on quality of care [41]. Given the diversity of their patients, HCs need SUD staff that are culturally and linguistically competent [41]. Recruiting and retaining such staff can be an additional challenge [41]. Similarly, challenges to increasing MAT capacity also exist. Physicians may require additional training and education to better serve SUD patients, particularly those with complex comorbid conditions who have a high rate of relapse [28]. Furthermore, having an on-site or nearby pharmacy to obtain the prescribed medication is critical to the success of delivery of MAT at HCs and help complete the continuum of comprehensive care [42]. In 2017, 43% of HCs had on-site pharmacy staff, but this number may need to grow [21, 44].

## Limitations

The limitations of our study include potentially underestimating SUD capacity because HCs have the option to report SUD services provided by mental health staff; we did not include mental health service provision in our analysis. Furthermore, SUD services are potentially undercounted because those provided by primary care or mental health providers are not included in UDS reporting and this bias is likely to underestimate any association between HRSA funding and the outcomes. We could not assess provision of SBIRT services by providers because HCs were not required to report this data in the UDS. Additionally, HRSA funds could have been used to employ or train staff, expand health information technology, or for other purposes in alignment with funding opportunities, but we lacked data on exactly how funds were used by HCs. We were unable to assess the proportion of national supply of MAT or SUD providers attributable to HCs, as providers can choose not to be listed in registries of providers with DATA waivers. Between 2016 and 2017, the definition of MAT provider was expanded beyond physicians to include certified nurse practitioners and physician assistants that were eligible to prescribe MAT for the treatment of Opioid Use Disorders, potentially overestimating our association between MAT capacity, service use, and panel size with receipt of HRSA supplemental funding. Furthermore, the most recently available UDS data was 2017, restricting our ability to assess the impact of the SASE and AIMS grants on SUD and MAT staffing, service use, provider panel size and visit ratio in later years. Our findings demonstrate that funding may promote colocation and increase in SUD and MAT capacity and service use are generalizable to HRSA-funded HCs but are also relevant to other health centers or primary care settings which focus on low-income and uninsured populations with a high burden of SUDs.

## Conclusions

Combined SASE and AIMS funding corresponded to colocation and increased SUD capacity and service use among HCs that received these grants. However, given the rapid escalation of the opioid epidemic and mortality rate, SUD and MAT capacity and service delivery by HCs has to continue to increase [8, 12, 13, 45]. Future research is needed to evaluate longer term outcomes of HRSA efforts to promote SUD and MAT capacity, how HCs invested these grants, and what best practices can be scaled up.

In 2018, HRSA awarded an additional $350 million to HCs through the funding opportunity Expanding Access to Quality Substance Use Disorder and Mental Health Services (SUD-MH) [46]. These funds are intended to implement and advance evidence-based strategies to expand access to integrated SUD and mental health services and reflect the continued commitment of HRSA in tackling the opioid epidemic and will likely lead to more growth in SUD and MAT capacity in HCs [19]. Colocation of SUD and MAT services is an important step in delivery of early intervention to avoid SUD morbidity and mortality and is crucial in battling the opioid epidemic. Initiating or expanding SUD capacity by other safety net providers might be challenging, but expanding MAT capacity among primary care physicians, in other settings, and increasing the patient cap to MAT providers are attainable and necessary national strategies to reduce opioid mortality [28, 47].

## Supporting information

**S1 Table. Full regression models of substance use disorder capacity, service use, and panel size and visit ratio by grantee status.**
(DOCX)

**S2 Table. Percent change and predicted probabilities of substance use disorder patients and visits by grantee status, from 2010 to 2017.**
(DOCX)

**S1 Dataset.**
(XLSX)

## Acknowledgments

**Disclaimer:** The views expressed in this publication are solely the opinions of the authors and do not necessarily reflect the official policies of the U.S. Department of Health and Human Services or the Health Resources and Services Administration, nor does mention of the department or agency names imply endorsement by the U.S. Government.

## Author Contributions

**Conceptualization:** Nadereh Pourat, Brenna O'Masta, Xiao Chen, Connie Lu, Weihao Zhou, Marlon Daniel, Hank Hoang, Alek Sripipatana.

**Data curation:** Brenna O'Masta, Connie Lu, Weihao Zhou.

**Formal analysis:** Xiao Chen, Weihao Zhou.

**Funding acquisition:** Nadereh Pourat.

**Investigation:** Nadereh Pourat.

**Methodology:** Nadereh Pourat, Brenna O'Masta, Xiao Chen.

**Project administration:** Connie Lu.

**Writing – original draft:** Nadereh Pourat, Brenna O'Masta, Connie Lu.

**Writing – review & editing:** Nadereh Pourat, Brenna O'Masta, Xiao Chen, Connie Lu, Weihao Zhou, Marlon Daniel, Hank Hoang, Alek Sripipatana.

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
