## [Decision Letter · Decision Letter 0]

17 Aug 2020

PONE-D-20-00610

Examining trends in substance use disorder capacity and service delivery by Health Resources and Services Administration-funded health centers: A time series analysis

PLOS ONE

Dear Dr. Pourat,

Thank you for submitting your manuscript to PLOS ONE. After careful consideration, we feel that it has merit but does not fully meet PLOS ONE’s publication criteria as it currently stands. Therefore, we invite you to submit a revised version of the manuscript that addresses the points raised during the review process.

Please revise the title to better reflect the nature of this study. Please explain what time series analysis methods were used. 

We look forward to receiving your revised manuscript.

Kind regards,

George Liu, PhD

Academic Editor

PLOS ONE

Journal Requirements:

Reviewers' comments:

Reviewer's Responses to Questions

**Comments to the Author**

1. Is the manuscript technically sound, and do the data support the conclusions?

Reviewer #1: Yes

Reviewer #2: Partly

2. Has the statistical analysis been performed appropriately and rigorously? 

Reviewer #1: Yes

Reviewer #2: Yes

3. Have the authors made all data underlying the findings in their manuscript fully available?

Reviewer #1: Yes

Reviewer #2: Yes

4. Is the manuscript presented in an intelligible fashion and written in standard English?

Reviewer #1: Yes

Reviewer #2: Yes

5. Review Comments to the Author

Reviewer #1: Congratulations on this work! This is a very useful paper; time trends have not been examined recently, and CHCs are such a critical part of the addiction treatment infrastructure in the US. I note some issues that could be clarified, and the takeaway messages should be sharpened. The gap between need and capacity could be presented more prominently (how many more FTE’s are needed?). After those corrections are made, this paper is a useful contribution to the literature.

Opioids are a high issue, but alcohol kills more people every year. When talking about SUD in the Intro, please also mention other drugs, at least once. You mention unmet need for addiction treatment in general, but the focus is squarely on opioids and it should not be. Meth is on the rise in many parts of the country, so we really don’t want to build opioid treatment capacity only—we want to build SUD treatment capacity. There is MAT for alcohol, but it is woefully underutilized.

One important issue that readers likely aren’t aware of is the fact that each health center operates multiple sites (6 or so, last time I checked). The should be stated clearly at the beginning of the paper, since when you say that 70% of CHCs have SUD capacity (line 107), this doesn’t mean that 70% of sites have capacity; it means that at least one site within 70% of grantees have capacity. This is a huge difference, and as currently written, the paper greatly overstates SUD capacity on the ground in health centers. Line 143 mentions that UDS data are at the grantee level, but not that most grantees operate multiple delivery sites. On line 19, the number of health center organizations is mentioned, and the number of care delivery sites should be included there as well.

The takeaway messages could be clearer in the abstract. In addition, this sentence in the abstract can be clarified: “From 2010 to 2015, 20% of health centers had any SUD staff, one full-time equivalent SUD staff

employed on average, and did not report a growth in SUD capacity or service delivery.”

One thing that should be discussed as a shortcoming is the way SUD capacity is measured: “(1) the proportion of HCs with at least one full-time equivalent (FTE) SUD staff, (2) the average number of SUD staff per HC, and (3) the ratio of SUD staff per 1,000 patients.” (line 180). So, all types of capacity aren’t captured, such as care provided by a MH specialist or primary care provider. This is a huge issue and should be described and made clear for the reader. Maybe the authors should say “specialist SUD capacity” instead of “SUD capacity.” This issue is mentioned in the Limitations, but attempts should be made to clarify this for the readers before they get to that point in the paper.

Was increasing SUD capacity a requirement of receiving the grants (the requirements should be described briefly)? If so, it’s surprising to readers that don’t consider the fact that a lot of SUD care is provided by non-SUD specialists that “Receipt of both supplementary grants increased the probability of any SUD capacity by 35%.” All the more reason to be clear on what is meant by capacity.

The service use measures might also be troubling. If they are from Table 6 in the UDS, those figures are commonly under-reported. If the numbers for “SUD patients” and “SUD visits” are just those patients served by SUD specialists, then this is an undercount since SUD patients/visits might be served by MH staff or primary care physicians. This shortcoming should be noted.

Line 123: a recent paper examines the impact of one of the grants: https://ps.psychiatryonline.org/doi/abs/10.1176/appi.ps.201900409

Line 122: Several newer studies are available that should be mentioned if the older and non-peer-reviewed papers are cited here (and I’d recommend removing the two non-peer-review cites, since they aren’t needed). These are just the ones that I know about, so rechecking the lit review on this point might make sense.

• Jones E. Medication-assisted opioid treatment prescribers in federally qualified health centers: capacity lags in rural areas. Journal of Rural Health. 2018;34(1):14-22

• Jones E, Rieckmann T. On-site mental health and substance use disorder screening and treatment capacity in health centers. Journal of Drug Issues. 2018;48(2):152-164.

• Jones E, Zur J, Rosenbaum S. Homeless caseload is associated with behavioral health and case management staffing in health centers. Administration and Policy in Mental Health and Mental Health Services Research. 2017;44(4):492-500.

• Jones E, Ku L. Sharing a playbook: integrated care in community health centers. American Journal of Public Health. 2015;105(10):2028-34.

• Jones E, Zur J, Rosenbaum S, Ku L. Opting out of Medicaid expansion: impact on encounters with behavioral health specialty staff in community health centers. Psychiatric Services. 2015;66(12):1277-82.

That’s neat that the authors used WONDER data too! I haven’t seen this merged with the UDS before. This doesn’t come across in the Abstract; I’d consider adding it there. I see NSSATS was also merged with the UDS, which might be mentioned in the abstract.

On average, Section 330 funding from HRSA only comprise 17% of each health center’s revenue (last time I checked). So, the “HRSA-funded” and “federally-funded” in the title and elsewhere should be eliminated, since it gives an outsize importance to HRSA funding.

Minor comments:

Line 106: it says self-reported data is likely an undercount, but the other data sources for SUD prevalence are largely self-report as well.

Line 173: Is PMCH recognition from the UDS? Also, it should be clarified that PCMH recognition could be at the site level and thus not cover all of a grantee’s sites.

I might have missed it, but I didn’t see a description of how the multivariable models were fitted.

Reviewer #2: PLOS One review Pourat et al

Examining trends in SUD capacity and service delivery by HRSA funded health centers: a time series analysis

This paper examines a critical topic, SUD staffing and service delivery by health centers. I believe work like this is critical to expand access to treatment beyond specialty care settings. In particular, the effects of HRSA’s SASE and AIMS grants are examined, which can be useful to guide future federal efforts to combat the opioid overdose epidemic.

Overall I feel the authors have done a good job with this paper, but two main concerns arose as I read it. These are detailed below.

• The SASE+AIMS group clearly outperformed the AIMS-only group, and the authors suggest AIMS simply hasn’t had time to reach maturity and they weren’t able to assess its full impact (p. 22). Both of these are reasonable suggestions, but it appears there may be more to it than that. It looks like the high-performing SASE+AIMS group was a fairly selective group (only 19% of the sample) that may have been qualitatively different. While I am not an expert in SASE and AIMS, my understanding is SASE was only available to FQHCS (recipients of section 330 grants), not FQHC look-alikes. If so, this appears to be an important confound to be acknowledged. AIMS, on the other hand, was not limited to FQHCs. As the authors acknowledge, at baseline the SASE+AIMS group had significantly more SUD staff, SUD visits at baseline, were bigger, etc. (Table 2, p. 15). In other words, they were better equipped to expand SUD treatment. These may have included FQHCs that either already had SUD treatment as part of their scope of project, or they added it as part of their participation in SASE (Form 5A of the application functioned as a Change in Scope request if SASE funding was received). By contrast AIMS did not require or include a change of scope. It appears, then, that the SASE grant would have provided its recipients with a springboard to increase SUD services because they could use the grant to expand these services, then (critically) they could sustain these services and staff beyond the end of the grant by billing for these services within their scope of project. For the AIMS-only group, on the other hand, a change of scope is explicitly not included, so the sustainability of the services may end along with the grant, providing them with far less incentive to expand. I am not disagreeing with the authors’ conclusions that SASE+AIMS was effective, but I think an important policy-relevant piece might be getting left out. It looks to me like the SASE grant likely helped, but it would be incorrect for a reader (and policymakers) to conclude the same results from SASE will necessarily occur with AIMS or other future funding. The lack of the same effect so far in the AIMS group seems to support this so far. The structure of SASE, particularly with respect to the scope of service and sustainability, may have been decisive. To be fair, the authors would have been criticized if they tried to make this point too strongly with the data available, but I would urge them to consider adding it to the discussion, limitations, and/or abstract if they agree it is likely. Right now as the paper stands I fear a reader of the abstract or conclusions (which are the only things some people read) would come away with the possibly oversimplified conclusion that all funding is inherently equally good funding. Having said all this, although I have done some work with FQHCs I do not consider myself an expert in FQHC financing so if HRSA or the authors feel I am off base on some part of this argument, maybe I am.

• The paper vacillates between discussing general SUD variables and opioid-related ones specifically, which can lead to confusion. Opioids are certainly a part of the SUD problem, but opioid trends can and often do diverge from those of other substances.

o For example, on p. 10 WONDER opioid mortality rates are presented as a measure for “substance abuse need”. Although the term “substance abuse need” is imprecise, I interpret it as shorthand for need for substance use disorder treatment. If this is correct, opioid mortality is not the same thing for a number of reasons, including the exclusion of other substances (cannabis, stimulants, alcohol and others all having their own trajectories) and changing lethality of substances over time (e.g. due to fentanyl), which can happen independently of treatment need as defined by SAMHSA (e.g. in their NSDUH survey). One fix would be to look for a better measure, but I don’t know of any readily available in annual form at the county level. The easiest fix is probably to simply say opioid mortality was used as a control, and not try to present it as a broader measure of need.

o Similarly, N-SSATS is used “to assess the supply of drug and alcohol treatment facilities in the county where the HC was located.” If the focus on opioids, a more precise control may be to count only treatment programs aimed at opioids (question 12 on the 2015 N-SSATS). However, again, some of the dependent variables in this paper are in fact for general SUD while others are specifically for opioids, which complicates things. One approach may be to use a different control variable for each DV, depending on the focus (SUD or OUD) though admittedly that would complicate the analyses substantially. Perhaps performing a sensitivity analysis, and just mentioning it in a footnote if it makes no difference, would be sufficient.

Minor items:

p. 21 “The growth in SUD capacity was timely and corresponded to increased service delivery, resulting in 0.11 SUD FTE staff per 1,000 patients in 2016, compared to national estimates of 0.07 outpatient SUD treatment staff per 1,000 patients.” “Delivery” is probably not the best term here, since staffing is being discussed rather than service delivery.

p. 23 “services provided by primary care providers are not included in UDS reporting.” What about SBIRT?

p. 24 “SASE and AIMS funding corresponded to colocation and increased SUD capacity and service use among HCs that received these grants.” Technically correct, but this could lead some readers to assume AIMS on its own has shown strong results. Consider rewording it to something along the lines of “combining SASE and AIMS funding . . . ”

6. PLOS authors have the option to publish the peer review history of their article (what does this mean?). If published, this will include your full peer review and any attached files.

Reviewer #1: No

Reviewer #2: **Yes: **Darren Urada

---

## [Author Response · Author response to Decision Letter 0]

25 Sep 2020

Reviewer #1

Reviewer Comment

Congratulations on this work! This is a very useful paper; time trends have not been examined recently, and CHCs are such a critical part of the addiction treatment infrastructure in the US. I note some issues that could be clarified, and the takeaway messages should be sharpened. The gap between need and capacity could be presented more prominently (how many more FTE’s are needed?). After those corrections are made, this paper is a useful contribution to the literature.

Response to Reviewers

Thank you for your comments. We have revised the paper in response to reviewer’s comments. 

Location in Manuscript

No change

Reviewer Comment

Opioids are a high issue, but alcohol kills more people every year. When talking about SUD in the Intro, please also mention other drugs, at least once. You mention unmet need for addiction treatment in general, but the focus is squarely on opioids and it should not be. Meth is on the rise in many parts of the country, so we really don’t want to build opioid treatment capacity only—we want to build SUD treatment capacity. There is MAT for alcohol, but it is woefully underutilized.

Response to Reviewers

We broadened this discussion as suggested. 

Location in Manuscript

Introduction, Page 6, line 140-145

Reviewer Comment

One important issue that readers likely aren’t aware of is the fact that each health center operates multiple sites (6 or so, last time I checked). The should be stated clearly at the beginning of the paper, since when you say that 70% of CHCs have SUD capacity (line 107), this doesn’t mean that 70% of sites have capacity; it means that at least one site within 70% of grantees have capacity. This is a huge difference, and as currently written, the paper greatly overstates SUD capacity on the ground in health centers. Line 143 mentions that UDS data are at the grantee level, but not that most grantees operate multiple delivery sites. On line 19, the number of health center organizations is mentioned, and the number of care delivery sites should be included there as well.

Response to Reviewers

The authors have clarified and included the number of delivery sites that are operated by health center organizations throughout the manuscript. 

Location in Manuscript

Introduction, Page 7, line 169; lines 176-177

Methods, Page 10, line 245

Throughout manuscript

Reviewer Comment

The takeaway messages could be clearer in the abstract. In addition, this sentence in the abstract can be clarified: “From 2010 to 2015, 20% of health centers had any SUD staff, one full-time equivalent SUD staff employed on average, and did not report a growth in SUD capacity or service delivery.”

Response to Reviewers

The authors have clarified this sentence. In addition, the authors have edited the Conclusions section of the Abstract. 

Location in Manuscript

Abstract, Page 4, line 61-63, line 69-122

Reviewer Comment

One thing that should be discussed as a shortcoming is the way SUD capacity is measured: “(1) the proportion of HCs with at least one full-time equivalent (FTE) SUD staff, (2) the average number of SUD staff per HC, and (3) the ratio of SUD staff per 1,000 patients.” (line 180). So, all types of capacity aren’t captured, such as care provided by a MH specialist or primary care provider. This is a huge issue and should be described and made clear for the reader. Maybe the authors should say “specialist SUD capacity” instead of “SUD capacity.” This issue is mentioned in the Limitations, but attempts should be made to clarify this for the readers before they get to that point in the paper.

Response to Reviewers

We included further clarification for focusing on SUD staff and inserted the word “specialized” several times to remind readers of the scope of this analysis. 

Location in Manuscript

Introduction, Page 9, line 228-230

Throughout manuscript

Reviewer Comment

Was increasing SUD capacity a requirement of receiving the grants (the requirements should be described briefly)? If so, it’s surprising to readers that don’t consider the fact that a lot of SUD care is provided by non-SUD specialists that “Receipt of both supplementary grants increased the probability of any SUD capacity by 35%.” All the more reason to be clear on what is meant by capacity.

Response to Reviewers

We have added additional detail on grant requirements to the text. These details indicate that screening, brief intervention, referrals, and care coordination, and collaboration with SUD providers were intended outcomes. We could not measure if SUD treatment by primary care providers increased, though we agree that primary care providers may have increased such activities.

We have added the term specialized SUD capacity when possible in response to this comment.

Location in Manuscript

Introduction, Page 8, line 189-195; line 199-204

Throughout manuscript

Reviewer Comment

The service use measures might also be troubling. If they are from Table 6 in the UDS, those figures are commonly under-reported. If the numbers for “SUD patients” and “SUD visits” are just those patients served by SUD specialists, then this is an undercount since SUD patients/visits might be served by MH staff or primary care physicians. This shortcoming should be noted.

Response to Reviewers

Our service use measures are from Table 5a. We had previously noted this in the Limitations section and expanded on this matter. 

Location in Manuscript

Limitations, Page 24, line 477-478

Reviewer Comment

Line 123: a recent paper examines the impact of one of the grants: 

https://ps.psychiatryonline.org/doi/abs/10.1176/appi.ps.201900409

Response to Reviewers

Thank you for this citation. We have included it in the Introduction. 

Location in Manuscript

Introduction, Page 9, line 217-218

Reviewer Comment

Line 122: Several newer studies are available that should be mentioned if the older and non-peer-reviewed papers are cited here (and I’d recommend removing the two non-peer-review cites, since they aren’t needed). These are just the ones that I know about, so rechecking the lit review on this point might make sense.

• Jones E. Medication-assisted opioid treatment prescribers in federally qualified health centers: capacity lags in rural areas. Journal of Rural Health. 2018;34(1):14-22

• Jones E, Rieckmann T. On-site mental health and substance use disorder screening and treatment capacity in health centers. Journal of Drug Issues. 2018;48(2):152-164.

• Jones E, Zur J, Rosenbaum S. Homeless caseload is associated with behavioral health and case management staffing in health centers. Administration and Policy in Mental Health and Mental Health Services Research. 2017;44(4):492-500.

• Jones E, Ku L. Sharing a playbook: integrated care in community health centers. American Journal of Public Health. 2015;105(10):2028-34.

• Jones E, Zur J, Rosenbaum S, Ku L. Opting out of Medicaid expansion: impact on encounters with behavioral health specialty staff in community health centers. Psychiatric Services. 2015;66(12):1277-82.

Response to Reviewers

We have updated our cited literature to more recent studies as suggested. 

Location in Manuscript

Introduction, Page 9, line 215

Reviewer Comment

That’s neat that the authors used WONDER data too! I haven’t seen this merged with the UDS before. This doesn’t come across in the Abstract; I’d consider adding it there. I see NSSATS was also merged with the UDS, which might be mentioned in the abstract.

Response to Reviewers

The authors have included in the Abstract mention of these two data sources. However, the name of these data sources were not included there due to word limits. 

Location in Manuscript

Abstract, Page 4, line 60-61

Reviewer Comment 

On average, Section 330 funding from HRSA only comprise 17% of each health center’s revenue (last time I checked). So, the “HRSA-funded” and “federally-funded” in the title and elsewhere should be eliminated, since it gives an outsize importance to HRSA funding.

Response to Reviewers

We have used the term “HRSA-funded” to highlight the fact that some health centers do not receive 330 funding. We agree with the reviewer that this amount is a small proportion of HC funding on average, but the proportion varies and it is higher for some HCs. It is also an important factor in the amount of care they can provide to uninsured patients. We have kept this language only in three locations.

Location in Manuscript 

Throughout manuscript

Reviewer Comment 

Minor comments: 

Line 106: it says self-reported data is likely an undercount, but the other data sources for SUD prevalence are largely self-report as well.

Response to Reviewers

The authors have acknowledged this point in the text. 

Location in Manuscript 

Introduction, Page 7, line 175-176

Reviewer Comment 

Line 173: Is PMCH recognition from the UDS? Also, it should be clarified that PCMH recognition could be at the site level and thus not cover all of a grantee’s sites.

Response to Reviewers

Yes, PCMH recognition is obtained from UDS data and the authors have mentioned this as the data source. The authors have also clarified that PCMH recognition can be obtained at the site level. 

Location in Manuscript 

Methods, Page 11, line 273-276

Reviewer Comment 

I might have missed it, but I didn’t see a description of how the multivariable models were fitted.

Response to Reviewers

We have described the needed information on how the models were fitted in the methods. However, we did not include fit statistics, because we are using random-effect models and cross-sectional longitudinal data. There are no readily available fit statistics for such models. However, in response to reviewer’s question, we examined the residuals, concordance, and correlation between predicted and observed values of our outcomes of interest manually. We found high correlation and high concordance between our predicted and observed values for all our outcome variables, indicating our models are appropriately fitted. 

Location in Manuscript 

Methods, Page 13, line 316

Reviewer #2

Reviewer Comment 

This paper examines a critical topic, SUD staffing and service delivery by health centers. I believe work like this is critical to expand access to treatment beyond specialty care settings. In particular, the effects of HRSA’s SASE and AIMS grants are examined, which can be useful to guide future federal efforts to combat the opioid overdose epidemic.

Overall I feel the authors have done a good job with this paper, but two main concerns arose as I read it. These are detailed below.

• The SASE+AIMS group clearly outperformed the AIMS-only group, and the authors suggest AIMS simply hasn’t had time to reach maturity and they weren’t able to assess its full impact (p. 22). Both of these are reasonable suggestions, but it appears there may be more to it than that. It looks like the high-performing SASE+AIMS group was a fairly selective group (only 19% of the sample) that may have been qualitatively different. While I am not an expert in SASE and AIMS, my understanding is SASE was only available to FQHCS (recipients of section 330 grants), not FQHC look-alikes. If so, this appears to be an important confound to be acknowledged. AIMS, on the other hand, was not limited to FQHCs. As the authors acknowledge, at baseline the SASE+AIMS group had significantly more SUD staff, SUD visits at baseline, were bigger, etc. (Table 2, p. 15). In other words, they were better equipped to expand SUD treatment. These may have included FQHCs that either already had SUD treatment as part of their scope of project, or they added it as part of their participation in SASE (Form 5A of the application functioned as a Change in Scope request if SASE funding was received). By contrast AIMS did not require or include a change of scope. It appears, then, that the SASE grant would have provided its recipients with a springboard to increase SUD services because they could use the grant to expand these services, then (critically) they could sustain these services and staff beyond the end of the grant by billing for these services within their scope of project. For the AIMS-only group, on the other hand, a change of scope is explicitly not included, so the sustainability of the services may end along with the grant, providing them with far less incentive to expand. I am not disagreeing with the authors’ conclusions that SASE+AIMS was effective, but I think an important policy-relevant piece might be getting left out. It looks to me like the SASE grant likely helped, but it would be incorrect for a reader (and policymakers) to conclude the same results from SASE will necessarily occur with AIMS or other future funding. The lack of the same effect so far in the AIMS group seems to support this so far. The structure of SASE, particularly with respect to the scope of service and sustainability, may have been decisive. To be fair, the authors would have been criticized if they tried to make this point too strongly with the data available, but I would urge them to consider adding it to the discussion, limitations, and/or abstract if they agree it is likely. Right now as the paper stands I fear a reader of the abstract or conclusions (which are the only things some people read) would come away with the possibly oversimplified conclusion that all funding is inherently equally good funding. Having said all this, although I have done some work with FQHCs I do not consider myself an expert in FQHC financing so if HRSA or the authors feel I am off base on some part of this argument, maybe I am.

Response to Reviewers

We agree with the reviewer and have acknowledged these points in the Discussion section. 

Please note that our data only included 330 grantees so our findings are not confounded by differences between HCs with and without 330 funding.

Location in Manuscript 

Discussion, Page 23, lines 439-443

Reviewer Comment 

• The paper vacillates between discussing general SUD variables and opioid-related ones specifically, which can lead to confusion. Opioids are certainly a part of the SUD problem, but opioid trends can and often do diverge from those of other substances.

o For example, on p. 10 WONDER opioid mortality rates are presented as a measure for “substance abuse need”. Although the term “substance abuse need” is imprecise, I interpret it as shorthand for need for substance use disorder treatment. If this is correct, opioid mortality is not the same thing for a number of reasons, including the exclusion of other substances (cannabis, stimulants, alcohol and others all having their own trajectories) and changing lethality of substances over time (e.g. due to fentanyl), which can happen independently of treatment need as defined by SAMHSA (e.g. in their NSDUH survey). One fix would be to look for a better measure, but I don’t know of any readily available in annual form at the county level. The easiest fix is probably to simply say opioid mortality was used as a control, and not try to present it as a broader measure of need.

Response to Reviewers

The authors have examined the available data and could not find publicly available data to measure substance use disorder mortality at the county level. We have reframed as the reviewer suggested to describe opioid mortality as a control measure. 

Location in Manuscript 

Methods, Page 11, line 278

Reviewer Comment 

o Similarly, N-SSATS is used “to assess the supply of drug and alcohol treatment facilities in the county where the HC was located.” If the focus on opioids, a more precise control may be to count only treatment programs aimed at opioids (question 12 on the 2015 N-SSATS). However, again, some of the dependent variables in this paper are in fact for general SUD while others are specifically for opioids, which complicates things. One approach may be to use a different control variable for each DV, depending on the focus (SUD or OUD) though admittedly that would complicate the analyses substantially. Perhaps performing a sensitivity analysis, and just mentioning it in a footnote if it makes no difference, would be sufficient.

Response to Reviewers

We agree that this control variable is not specific to opioid treatment supply. We tested using the variable suggested by the reviewer and found our main findings to be unchanged. We did not replace our original variable because the opioid treatment programs are aggregated at the state-level which is less precise than our measure of supply. Therefore, we kept the original variable in the models. 

Location in Manuscript 

No change 

Reviewer Comment 

Minor items: 

p. 21 “The growth in SUD capacity was timely and corresponded to increased service delivery, resulting in 0.11 SUD FTE staff per 1,000 patients in 2016, compared to national estimates of 0.07 outpatient SUD treatment staff per 1,000 patients.” “Delivery” is probably not the best term here, since staffing is being discussed rather than service delivery.

Response to Reviewers

We have revised this sentence for clarity. 

Location in Manuscript 

Discussion, Page 22, line 417

Reviewer Comment 

p. 23 “services provided by primary care providers are not included in UDS reporting.” What about SBIRT?

Response to Reviewers

HCs report SBIRT overall in Table 6 but do not indicate who conducted the screening or intervention. So we did not examine SBIRT. We have added this limitation.

Location in Manuscript 

Limitations, Page 25, line 480-481

Reviewer Comment 

p. 24 “SASE and AIMS funding corresponded to colocation and increased SUD capacity and service use among HCs that received these grants.” Technically correct, but this could lead some readers to assume AIMS on its own has shown strong results. Consider rewording it to something along the lines of “combining SASE and AIMS funding . . . ”

Response to Reviewers

We have revised this sentence as suggested. 

Location in Manuscript 

Conclusions, Page 25, line 497

Academic Editor

Reviewer Comment 

Please revise the title to better reflect the nature of this study. Please explain what time series analysis methods were used. 

Response to Reviewers

We have edited our title to include our methodology and clarified our methods in the Methods section.

Location in Manuscript 

Title

Methods, Page 13, line 316

---

## [Decision Letter · Decision Letter 1]

3 Nov 2020

Examining trends in substance use disorder capacity and service delivery by Health Resources and Services Administration-funded health centers: A time series regression analysis

PONE-D-20-00610R1

Dear Dr. Pourat,

We’re pleased to inform you that your manuscript has been judged scientifically suitable for publication and will be formally accepted for publication once it meets all outstanding technical requirements.

Kind regards,

George Liu, PhD

Academic Editor

PLOS ONE

Additional Editor Comments (optional):

Reviewers' comments:

Reviewer's Responses to Questions

**Comments to the Author**

1. If the authors have adequately addressed your comments raised in a previous round of review and you feel that this manuscript is now acceptable for publication, you may indicate that here to bypass the “Comments to the Author” section, enter your conflict of interest statement in the “Confidential to Editor” section, and submit your "Accept" recommendation.

Reviewer #1: All comments have been addressed

Reviewer #2: All comments have been addressed

2. Is the manuscript technically sound, and do the data support the conclusions?

Reviewer #1: Yes

Reviewer #2: Yes

3. Has the statistical analysis been performed appropriately and rigorously? 

Reviewer #1: Yes

Reviewer #2: Yes

4. Have the authors made all data underlying the findings in their manuscript fully available?

Reviewer #1: Yes

Reviewer #2: Yes

5. Is the manuscript presented in an intelligible fashion and written in standard English?

Reviewer #1: Yes

Reviewer #2: Yes

6. Review Comments to the Author

Reviewer #1: Great job addressing the comments! I have no further comments.

Reviewer #2: The authors have done a good job addressing previous comments. I am satisfied with their responses.

7. PLOS authors have the option to publish the peer review history of their article (what does this mean?). If published, this will include your full peer review and any attached files.

Reviewer #1: No

Reviewer #2: **Yes: **Darren Urada

---

## [Editor Report · Acceptance letter]

17 Nov 2020

PONE-D-20-00610R1 

Examining trends in substance use disorder capacity and service delivery by Health Resources and Services Administration-funded health centers: A time series regression analysis   

Dear Dr. Pourat:

I'm pleased to inform you that your manuscript has been deemed suitable for publication in PLOS ONE. Congratulations! Your manuscript is now with our production department. 

Kind regards, 

on behalf of

Dr. George Liu 

Academic Editor

PLOS ONE